# ExpressAnalyst: A unified platform for RNA-sequencing analysis in non-model species

Peng Liu[1,2], Jessica Ewald [1,2], Zhiqiang Pang[1], Elena Legrand[1], Yeon Seon Jeon [1], Jonathan Sangiovanni[1], Orcun Hacariz[1], Guangyan Zhou[1], Jessica A. Head[1], Niladri Basu[1] & Jianguo Xia [1] ✉

The increasing application of RNA sequencing to study non-model species demands easy-to-use and efficient bioinformatics tools to help researchers quickly uncover biological and functional insights. We developed ExpressAnalyst (www.expressanalyst.ca), a web-based platform for processing, analyzing, and interpreting RNA-sequencing data from any eukaryotic species. ExpressAnalyst contains a series of modules that cover from processing and annotation of FASTQ files to statistical and functional analysis of count tables or gene lists. All modules are integrated with EcoOmicsDB, an ortholog database that enables comprehensive analysis for species without a reference transcriptome. By coupling ultra-fast read mapping algorithms with high-resolution ortholog databases through a user-friendly web interface, ExpressAnalyst allows researchers to obtain global expression profiles and gene-level insights from raw RNA-sequencing reads within 24 h. Here, we present ExpressAnalyst and demonstrate its utility with a case study of RNA-sequencing data from multiple non-model salamander species, including two that do not have a reference transcriptome.

The last decade has seen growing applications of RNA-seq to environmental and agricultural studies involving non-model organisms[1]. Reference genomes/transcriptomes are not available for many of these species, and thus de novo-assembled transcriptomes are typically required to quantify raw RNA-seq reads. The approach contains two main steps: transcript assembly and gene annotation. The first step, assembly, involves piecing together putative transcripts from the raw RNA-seq data, which is a computationally intensive task that typically requires weeks of runtime on high-performance computers (HPC) equipped with 100 s of gigabytes (GB) of memory. Commonly used software includes Trinity, SOAPdenovo-Trans, and Oases[2–4]. In the second step, possible annotations are assigned to the assembled transcripts in the form of gene symbols, short descriptions, and functions as defined by several widely used pathway libraries and ontologies such as Kyoto Encyclopedia of Genes and Genomes (KEGG) and Gene Ontology (GO)[5,6]. Annotation is usually performed with Basic Local Alignment Search Tool (BLAST) like algorithms through a process called

"annotation transfer" that leverages existing genome annotations from other species[7]. Despite the wide adoption of this strategy, there is no consensus approach for choosing how many or which genomes to use, or for how to resolve multiple, conflicting, or missing annotations[8]. Count tables obtained from those de novo-assembled transcriptomes are difficult to analyze and interpret, as this process often results in hundreds of thousands of transcripts, most of which have either inconsistent or missing functional annotations (i.e., hypothetical proteins)[8,9]. In summary, the current practice for RNA-seq analysis involving non-model species is computationally intensive, requires advanced programming skills, and produces transcript IDs (usually in-house IDs) that are difficult to reproduce, compare across studies, and ultimately re-use. There is an urgent demand for computationally efficient, user-friendly, reproducible, and functionally coherent methods for RNA-seq processing and analysis for non-model species[10–12].

We previously developed a command-line algorithm, Seq2Fun for mapping RNA-seq reads from eukaryotic species to KEGG ortholog

---

[1]Faculty of Agricultural and Environmental Sciences, McGill University, Ste-Anne-de-Bellevue, Canada. [2]These authors contributed equally: Peng Liu, Jessica Ewald. ✉e-mail: jeff.xia@mcgill.ca

(KO) databases with a translated search[13]. We demonstrated that Seq2Fun outperforms de novo assembly in terms of accuracy, precision, computing time, and memory usage, while producing functional profiles that are consistent with traditional methods[13]. Despite this promising result, subsequent testing revealed several important limitations associated with this KO-based annotation system. The first is limited transcriptome coverage. Not all protein-coding genes are annotated with KOs. For example, the human genome has 19,648 protein-coding genes, of which only 14,964 (76.16%) are annotated with KOs[14]. Coverage is especially lower for non-mammalian species. For example, the zebrafish genome has 26,584 protein-coding genes and only 16,322 (61.40%) are annotated with KOs. This incomplete transcriptome coverage is biased, particularly for non-mammalian species for which fewer KEGG pathways are defined, and some biological processes, such as egg yolk formation in oviparous species, are notably absent. For example, vitellogenin, a precursor to egg yolk protein formation and a biomarker of interest[15], is not found in the KO system. The second is limited transcript resolution as KO groups often gather many genes from one species together. While this is inevitable to some extent during ortholog definition, increased resolution would be an asset to the user community. The third is limited functional annotation beyond KEGG pathways. Finally, the command line interface of Seq2Fun represents a significant barrier to researchers with limited programming skills.

Here, we describe a unified conceptual framework for reads quantification and annotation of RNA-seq data from non-model species and present its implementation in ExpressAnalyst, a web-based platform to streamline the whole process (Fig. 1). By leveraging the concept that short reads (~75 bp) can be uniquely mapped to ortholog groups[10,11], the typical steps of transcript assembly and annotation are replaced with the alignment of individual translated reads directly to a comprehensive, high-resolution protein ortholog database created from the genomes of hundreds of eukaryotic species. Downstream analysis is performed on the resulting high-resolution ortholog count table, similar to increasingly popular strategies to process shotgun metagenomics data[12]. To demonstrate the utility of ExpressAnalyst, we provide a case study involving RNA-seq data from three species

of salamanders. In doing so we show that researchers can use ExpressAnalyst to obtain comprehensive functional insights from raw RNA-seq reads from any eukaryotic species generally in less than 24 h of runtime, the majority of which is unsupervised data upload (if using our public server) and processing time. We have also developed a Docker version of this module that can be run on personal computers by individuals who have limited bandwidth or privacy concerns associated with raw sequence data.

## Results

### Overview of the software ecosystem

ExpressAnalyst supports comprehensive RNA-seq analysis from raw reads processing to statistical and functional analysis for any eukaryotic species. For species without a reference transcriptome, this is achieved by using the Seq2Fun algorithm to map reads to a comprehensive ortholog database – EcoOmicsDB (described below). While the primary motivation for ExpressAnalyst is to improve RNA-seq analysis for non-model organisms, it can also be used to analyze data from common model organisms. The main components of ExpressAnalyst are described in more detail below.

### EcoOmicsDB: A high-resolution ortholog database

EcoOmicsDB is a custom ortholog database that was developed to significantly improve the resolution and transcriptome coverage of the KEGG ortholog database used by Seq2Fun version 1.0[13]. It currently includes ~13 million protein-coding genes from 687 species (Table 1). Of these protein-coding sequences, 5,871,017 were annotated with KEGG pathways and 1,567,627 were annotated with GO terms. These 687 species were organized into 29 taxonomic sub-groups, based on the NCBI taxonomy[16]. Symbols, descriptions, and functional annotations were harmonized across individual proteins for each ortholog group (more details are provided in the methods section). All details for each ortholog group in EcoOmicsDB are accessible via ExpressAnalyst (https://expressanalyst.ca/EcoOmicsDB/), and can be queried by either ortholog ID or Entrez ID.

EcoOmicsDB was created with the OrthoFinder software[17], which identifies rooted ortholog groups by inferring groups of sequences

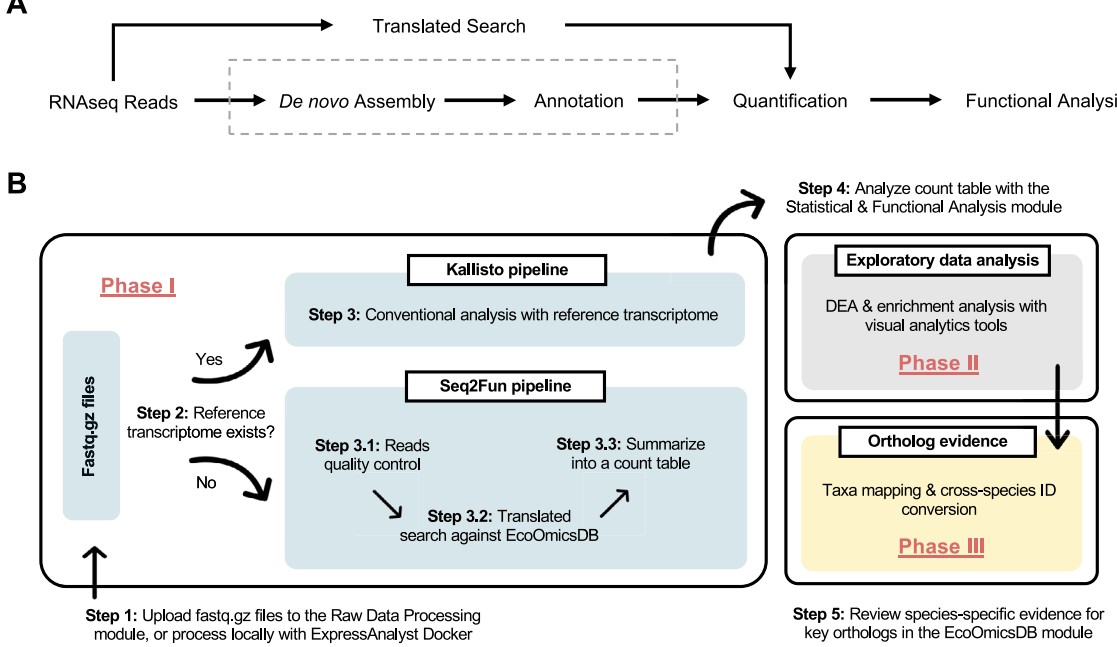

**Fig. 1 | Concept for and implementation of ExpressAnalyst. A** Conceptual solution to bypass the computationally intensive steps of de novo assembly and annotation. **B** Practical implementation of the conceptual solution in ExpressAnalyst. DEA Differential expression analysis.

**Table 1 | EcoOmicsDB contains 29 taxonomic sub-groups available for RNA-seq annotation and quantification of non-model organisms in eukaryotes**

| Level | Group | Species | Proteins | Ortholog | Seq2Fun (v1) |
|---|---|---|---|---|---|
| 1 | **Eukaryotes** | 687 | 12,828,537 | 666,067 | 8041 |
| 2 | <u>Animals</u> | 370 | 7,150,735 | 270,089 | - |
| 3 | *Vertebrates* | 212 | 4,588,985 | 83,704 | 6723 |
| 4 | Mammals | 94 | 1,910,363 | 47,144 | 5883 |
| 4 | Birds | 31 | 482,205 | 22,397 | 4177 |
| 4 | Reptiles | 20 | 384,584 | 21,725 | 4342 |
| 4 | Fishes | 61 | 1,736,572 | 42,497 | 4308 |
| 3 | *Arthropods* | 119 | 1,727,651 | 113,673 | 3541 |
| 4 | Insects | 101 | 1,376,824 | 70,170 | - |
| 4 | Crustaceans | 7 | 154,960 | 37,407 | - |
| 3 | *Cnidarians* | 9 | 203,000 | 24,003 | - |
| 3 | *Mollusks* | 9 | 206,905 | 35,775 | - |
| 3 | *Nematodes* | 6 | 134,093 | 35,865 | 2324 |
| 2 | <u>Plants</u> | 127 | 3,968,027 | 162,988 | 3012 |
| 3 | *Eudicots* | 93 | 3,180,221 | 102,677 | - |
| 3 | *Monocots* | 17 | 560,027 | 43,451 | - |
| 3 | *Algae* | 14 | 155,495 | 38,334 | - |
| 2 | <u>Fungi</u> | 138 | 1,278,312 | 148,080 | 2423 |
| 3 | *Ascomycetes* | 100 | 904,642 | 98,151 | - |
| 4 | Eurotiomycetes | 20 | 196,228 | 25,710 | - |
| 4 | Saccharomycetes | 36 | 195,913 | 14,873 | - |
| 4 | Dothideomycetes | 10 | 123,200 | 28,898 | - |
| 3 | *Basidiomycetes* | 33 | 363,997 | 56,935 | - |
| 2 | <u>Protists</u> | 52 | 660,237 | 134,451 | 2696 |
| 3 | *Alveolates* | 21 | 207,674 | 51,205 | - |
| 4 | Apicomplexans | 18 | 93,576 | 14,632 | - |
| 3 | *Stramenopiles* | 8 | 119,746 | 31,581 | - |
| 3 | *Amoebozoa* | 7 | 81,844 | 22,114 | - |
| 3 | *Euglenozoa* | 9 | 86,483 | 12,363 | - |

The taxonomic levels are indicated by the number in the Level column and also by indentations and font styles in the Group column (level 1 = bold, level 2 = underline, level 3 = italics, level 4 = regular). The number of orthologs in the KO database used by Seq2Fun version 1.0 are shown in the last column.

that share a common ancestor. The sequence-similarity parameters for ortholog definition were chosen to produce ortholog groups at a higher resolution than the KO system, and species-specific functional annotations were compiled to produce both KEGG and GO term gene sets for Seq2Fun ortholog IDs[5,14]. The first round of analysis took more than ten days to complete using a server with 54-threads and 504GB RAM. The analysis binned 12,828,537 protein-coding sequences from 687 species into 666,067 ortholog groups. The size distribution of these ortholog groups largely follow a power law distribution (Fig. 2A)[18,19]. While most orthologs group contain fewer than ten sequences, the largest ones were many times larger than this, with the biggest one (s2f_0000000) containing more than 50,000 transcription factor sequences. Aggregating at this level makes it difficult to infer gene-level insights. For example, our collaborators in ecotoxicology are particularly concerned by the 5[th] largest ortholog group (s2f_0000004) that contained >28,000 cytochrome P450 enzymes, an average of 47 per species. To address this issue, we further split the largest 10, 000 ortholog groups into 76, 066 groups with an adaptive k-means clustering-based approach (Fig. 2B), with the largest ortholog group being split into more groups ($n = 96$) than the smallest ($n = 2$). An example of this is shown for the vitellogenin ortholog group

(Fig. 2C), a protein family that is important in the study of non-mammalian vertebrate species because it is a highly sensitive bio-marker for exposure to estrogenic compounds. It is not found in the KEGG ortholog database. Details on the ortholog splitting approach, parameters, and rationale are given in the Methods section.

**Efficient FASTQ processing via web interface**

ExpressAnalyst contains a raw data processing module to produce count tables from raw RNA-seq reads for any eukaryotic species. For species with no reference transcriptome, reads are aligned to EcoO-micsDB using Seq2Fun 2.0. For species with a reference transcriptome, users can choose between using either Seq2Fun or Kallisto[20]. ExpressAnalyst has a user account system to allow users to upload, store, and process FASTQ files on our server. Each account is limited to 30GB of data. If users have data privacy concerns, have a dataset larger than 30 GB, or want to avoid the time-consuming upload step, we provide a Docker image for local installation. A detailed description of the overall raw data processing workflow is given in the Supplementary Note 1 and Supplementary Fig. 1.

ExpressAnalyst uses version 2.0 of the Seq2Fun algorithm (www. seq2fun.ca). Version 2.0 significantly reduces memory footprint while maintains a high efficiency compared to version 1.0[13]. For example, Seq2Fun 2.0 maintains ~2 million reads per minute while decreasing memory usage from 1.49 GB to 0.94 GB in processing our test datasets, despite version 2.0 having a much larger (>125 times increase) data-base compared to version 1.0. Version 2.0 also includes a new function called SeqTract to retrieve the mapped reads for a given list of genes for transcript assembly. Seq2Fun generates mapped reads of all genes into a single fastq.gz or a pair of fastq.gz files for single and pair-end reads, respectively. To conduct a target gene assembly, SeqTract takes a list of Seq2Fun IDs and the mapped fastq.gz files as an input and outputs a fastq.gz file for each ID. The file can be fed into popular de novo assemblers[21] to assemble the contigs for primer design, isoform identification, or phylogenetic analysis.

**Statistical and functional analysis of gene expression data**

ExpressAnalyst supports visualization, statistics, and functional ana-lysis of gene expression data uploaded in list or table format. There are three modules for statistical analysis according to the data input type. Here, we focus on the gene expression table module which contains many new and updated features to specifically support results gener-ated by the Seq2Fun algorithm.

The 'Expression Profiling' module accepts microarray or RNA-seq tables for 28 model species, as well as Seq2Fun IDs and KEGG ortholog IDs for any eukaryotic species with transcriptomics data. Additionally, users can also upload their own custom annotation files or sepa-rate metadata files containing multiple experimental factors. After data upload, the 'Data Quality Check' page displays textual and visual summaries of the transcriptomics data and metadata. Next, the 'Data Filtering & Normalization' page allows users to filter by variance, abundance, and perform several common normalization techniques. Boxplots, density plots, and principal component analysis (PCA) plots allow the users to examine their data before and after applying these normalization steps. On the 'Differential Expression Analysis' (DEA) page, users can select between several established DEA methods – limma, edgeR and DESeq2[22–24] with their associated parameters. Next is the 'Select Significant Genes' page, where users can define their p-value and/or fold-change cut-offs and view the gene expression results (Fig. 3A). All genes in the table are linked to their NCBI gene cards or EcoOmicsDB ortholog profiles (Fig. 3B). Users can view violin plots of expression values across experimental factors (Fig. 3A). Finally, the 'Analysis Overview' provides six visual analytics functions to help identify important features, functions and their correlations through interactive Volcano Plot, Enrichment Network, Ridgeline Chart, Heat-map, etc. (Fig. 3C, D).

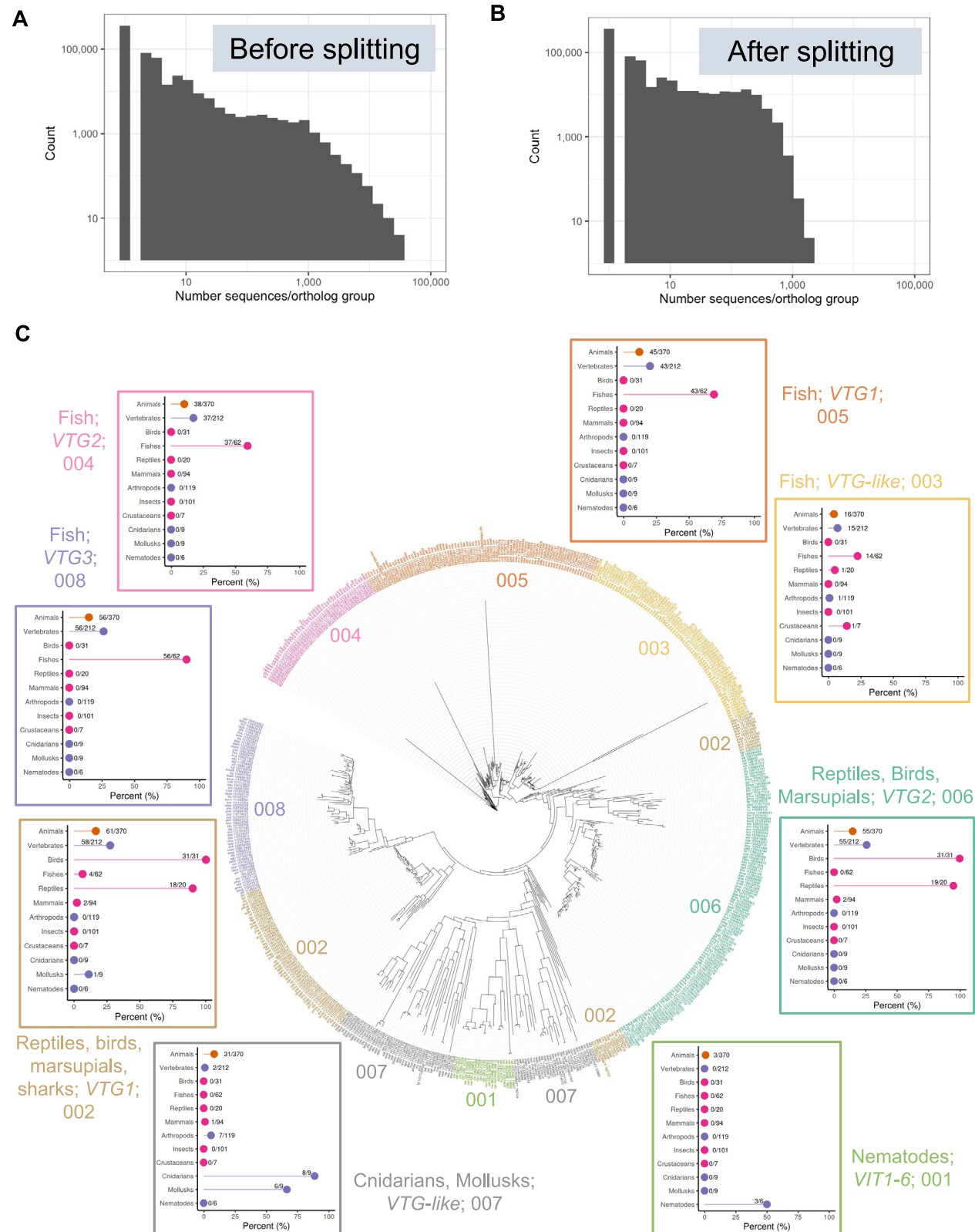

**Fig. 2 | Ortholog refinement in EcoOmicsDB.** Histogram of ortholog size before (**A**) and after (**B**) ortholog splitting. **C** Phylogenetic tree of sequences within ortholog s2f_0005567 (vitellogenin), annotated with the eight clusters created during ortholog splitting, surrounded by species hit plots for each cluster. Clusters are annotated by both distinct colors and numerical IDs (light green = 001,

brown = 002, yellow = 003, pink = 004, orange = 005, dark green = 006, grey = 007, purple = 008). The y-axis of species hit plots shows taxonomic groups within the animal kingdom (orange = level 1, purple = level 2, pink = level 3); and the x-axis shows the percentage of those species in EcoOmicsDB with a sequence in the ortholog group.

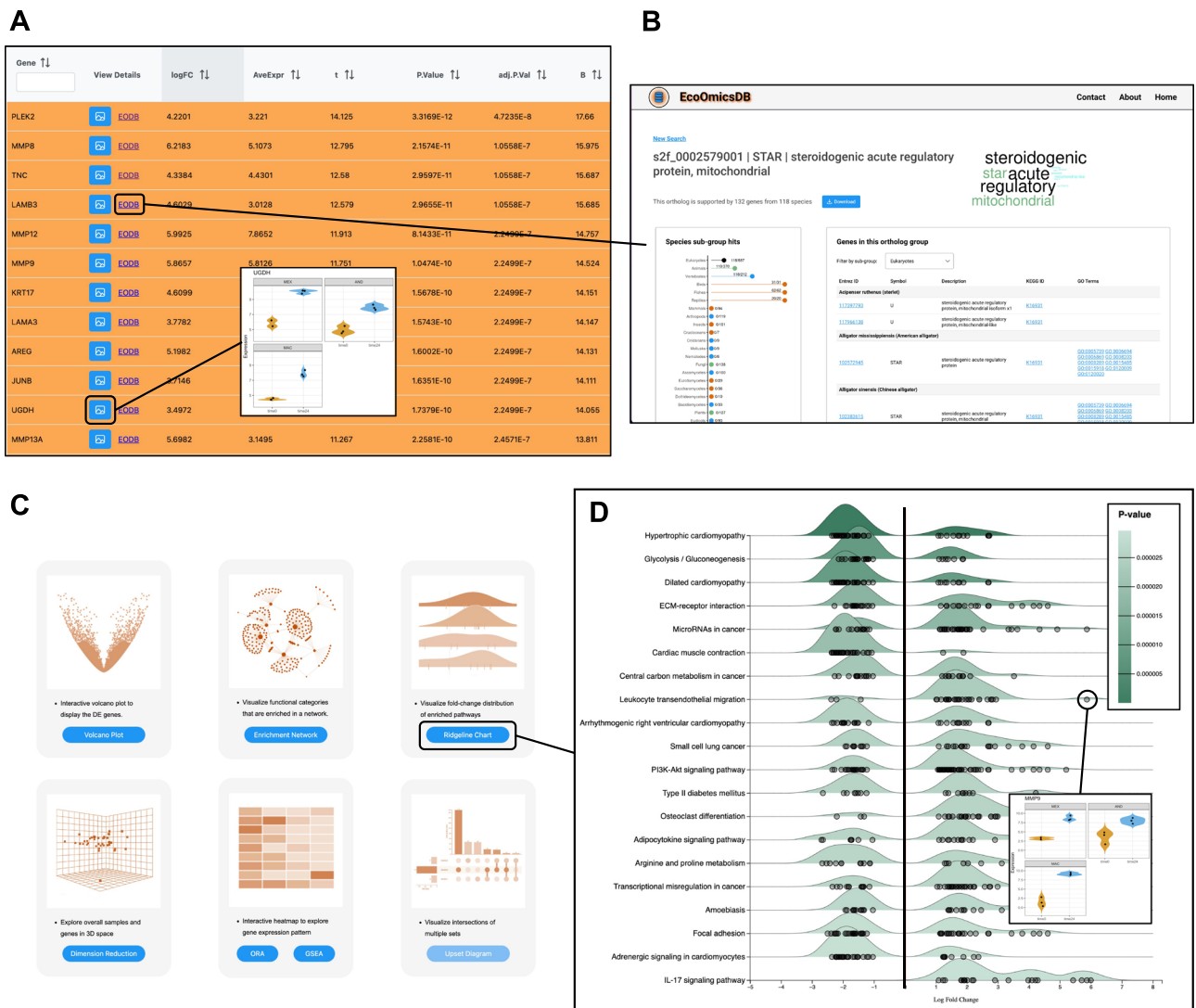

**Fig. 3 | Differential and functional analysis with ExpressAnalyst.** Key features of the ExpressAnalyst 'Expression Profiling' module include an interactive table of differentially expressed genes (**A**), with links to EcoOmicsDB ortholog profiles (**B**). There are six interactive visual analytics tools (**C**), one of which is the ridgeline plot (**D**). Here, the ridgeline plot shows gene set enrichment analysis results, where *p*-values were calculated based on the GSEA algorithm as implemented in the fgsea R package. In violin plots in (**A**) and (**D**), orange = time0 and blue = time24.

## Benchmarking and case studies

Seq2Fun 1.0 was rigorously validated in our original publication[13]. Here, to ensure that Seq2Fun 2.0 also reproduces results obtained using traditional approaches, we carried out two cases studies using organisms with reference transcriptomes (American lobster and zebrafish), as well as with one new case study involving salamander species with and without reference genomes. Seq2Fun 2.0 produced almost identical PCA variance structures and relative numbers of DEGs between treatment groups compared to analysis with reference transcriptomes (Supplementary Note 2, Supplementary Figs. 2, 3, Supplementary Tables 1, 2, Supplementary Data 1 and 2), following results obtained for Seq2Fun 1.0. Here, we focus on the third case study to demonstrate how the concepts and functions described in this paper can be used to efficiently analyze and interpret a comparative transcriptomics dataset from multiple salamander species, some without reference transcriptomes.

The RNA-seq dataset was originally collected as part of a comparative study of transcriptional responses to limb regeneration in three ambystomatid salamander species[25], one with a reference genome (*Ambystomatidae mexicanum*, abbreviation MEX) and two without (*Ambystomatidae andersoni*, abbreviation AND; *Ambystomatidae*

*maculatum*, abbreviation MAC). In the original experiment, an upper arm was amputated from larvae from each of the three species, and tissue samples were taken at the time of amputation (time0), and 24 h after amputation (time24). Three pools of five larvae from each species and time group were sequenced, resulting in 3 reps * 2 time points * 3 species = 18 RNA-seq samples. RNA-seq data was quantified using a reference transcriptome from MEX and de novo transcriptomes from AND and MAC. Differential expression analysis was conducted separately for each species, and differentially expressed genes (DEGs) that were shared across species were identified by searching for sequence similarities using the BLAST algorithm. Details on larvae sex were not provided by the original manuscript.

For our analysis, FASTQ files were downloaded from NCBI GEO and were re-processed with the Seq2Fun algorithm (vertebrates database) using ExpressAnalyst to obtain the gene count tables. Downstream statistical and functional analysis was performed with the 'Expression Profiling' module in ExpressAnalyst. PCA plots on the "Data Normalization" page show that the primary source of variability in the normalized count matrix is species differences, as shown by the separation patterns according to species along PC1 (Fig. 4A). AND and MEX samples fall closer to each other than to

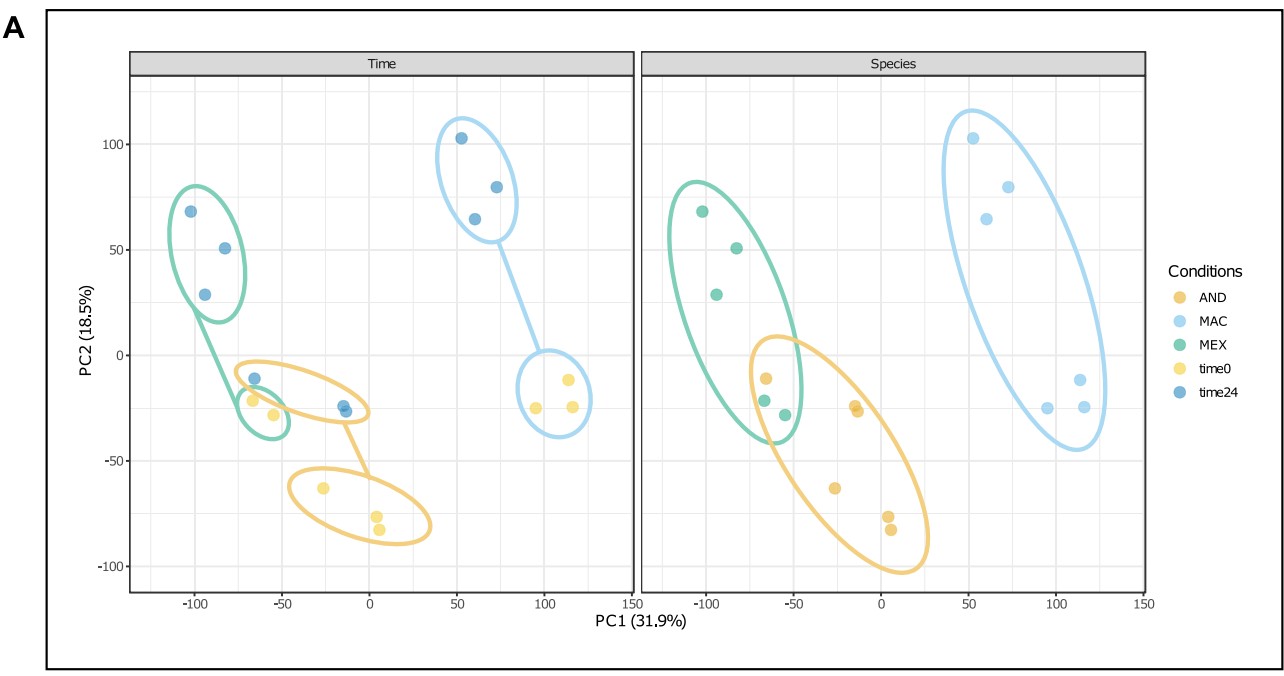

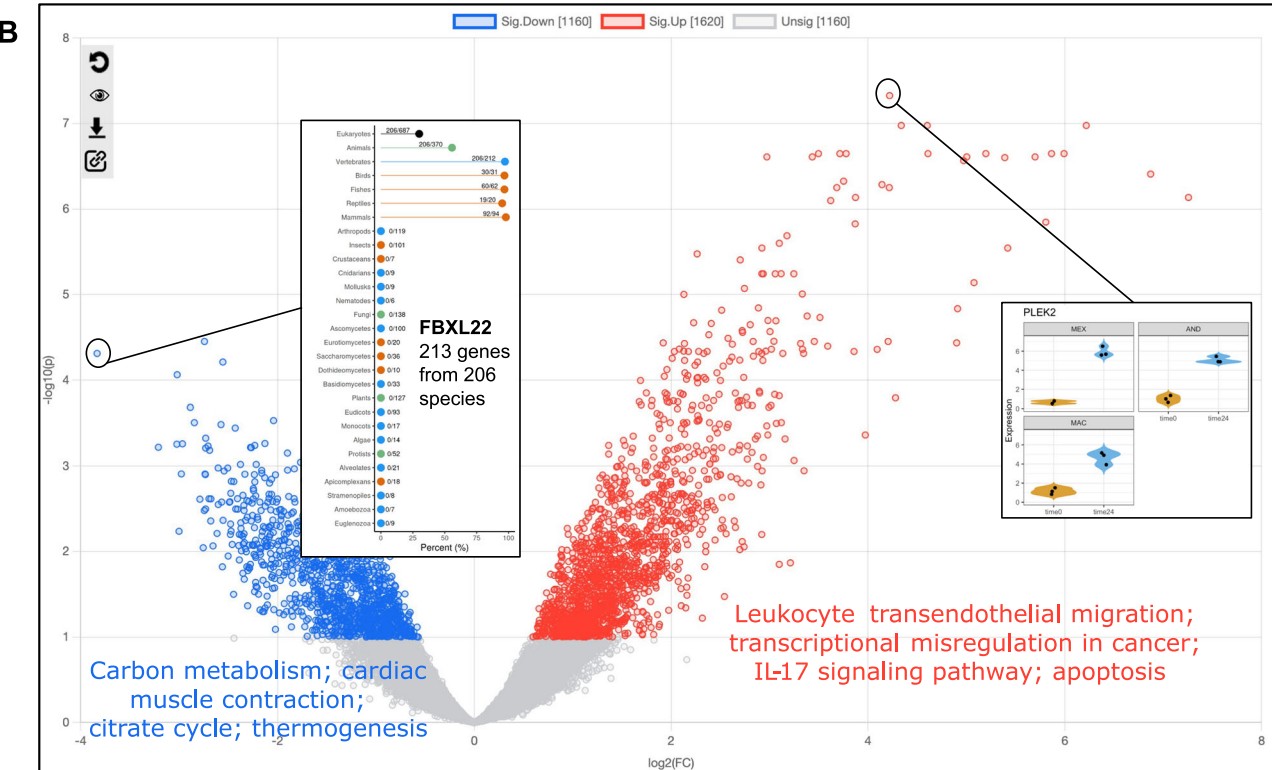

**Fig. 4 | Non-model organism case study results. A** PCA of normalized count tables for the salamander data, with samples annotated by time since amputation (time0 = yellow, time24 = dark blue), and species (AND = orange, MAC = light blue, MEX = green). **B** Volcano plot showing differentially expressed genes with log2FC on the x-axis and unadjusted *p*-value on the y-axis. Genes are colored red if their adjusted *p*-values < 0.05 and log2FCs > 0 and blue if their adjusted *p*-values < 0.05 and log2FCs < 0. *P*-values were calculated with the limma R package, which uses a moderated t-statistic (two-sided). Gene-specific details are displayed for PLEK2 (violin plot; time0 = orange, time24 = blue) and FBXL22 (species-hits plot). All plots were generated by ExpressAnalyst.

MAC, which is consistent with the differences in estimated divergence time − 4.27 million years between AND and MEX compared to 21.47 million years between AND and MAC[26]. The second largest source of variability was time since amputation, shown by the separation of samples with time0 and time24 annotations along PC2 (Fig. 4A).

Differential expression analysis was performed with the *limma* option to identify genes that were significantly different between time0 and time24 across species by analyzing all samples together and considering species as a blocking factor. This means that a linear mixed-effects model was fit to the expression profile of each gene, with species included as a random effect and time as a fixed effect. Then,

**Table 2 | Top five DEGs from the salamander case study**

| Seq2Fun ID | Symbol | Adj. p-value | log2FC | # Entrez ID | # Species |
|---|---|---|---|---|---|
| s2f_0005954003 | *PLEK2* | 4.72E -8 | 4.22 | 222 | 207 |
| s2f_0000105017 | *MMP8* | 1.06E −7 | 6.22 | 156 | 143 |
| s2f_0000040013 | *TNC* | 1.06E −7 | 4.34 | 309 | 218 |
| s2f_0000131002 | *LAMB3* | 1.06E −7 | 4.60 | 234 | 211 |
| s2f_0000105021 | *MMP12* | 2.25E -7 | 5.99 | 463 | 153 |

The *p*-value and log2FC statistics are from the ExpressAnalyst differential expression analysis results which uses a moderated t-statistic to calculate a two-sided *p*-value; the # Entrez ID and # Species are from EcoOmicsDB ortholog profiles.

the specific contrast between time0 and time24 was computed, which is the same thing as performing a two-sided moderated t-test. The mixed effects model takes variability associated with species into account when calculating the p-value for differences between time0 and time24. Using the same statistical thresholds as the original publication (adj. *p*-value < 0.1 (FDR), no log2FC cut-off), a total of 2780 DEGs were identified. The complete results from differential analysis are available in Supplementary Data File 3.

The "Interactive Volcano Plot" tool in ExpressAnalyst was used to perform over-representation analysis (ORA) separately on the up- and down-regulated genes with KEGG, Gene Ontology Biological Process (GO BP), Molecular Function (GO MF), and Cellular Component (GO CC) gene sets (Fig. 4B). Overall, there were 27 significantly enriched pathways in the list of up-regulated genes and 81 in the list of down-regulated genes. The up-regulated pathways were mainly related to immune response, cell proliferation, and programmed cell death. The down-regulated pathways were mainly related to muscle tissue and cellular metabolism. This is consistent with the functional analysis results reported by the original publication which noted that up and down-regulated genes were enriched in GO BP terms related to wound healing and tissue development, and muscle tissue and cellular metabolism, respectively[25]. ORA results are available in Supplementary Data 3.

The top five DEGs (Table 2) were queried against EcoOmicsDB to investigate gene-level details. For convenience, the table of differential expression results in the 'Expression Profiling' module contains hyperlinks to the EcoOmicsDB profiles for all Seq2Fun ortholog IDs. Further, graphics on the coverage of different species sub-groups provide valuable insights into the taxonomic domain of Seq2Fun ortholog groups.

The EcoOmicsDB profiles show that the top five DEGs are supported by lots of evidence (> 155 genes and > 142 species for each). The top four orthologs have an Entrez ID to species ratio close to one, and examination of the tables in EcoOmicsDB show that in cases where there are multiple Entrez IDs from one species, the symbols and descriptions are very similar (i.e., *TNC* and *TNC*-like isoform). The 5th ortholog (s2f_0000105021) contains on average three Entrez ID sequences per species. Examination of the EcoOmicsDB output shows that these are generally matrix metalloproteinase-3 (also known as stromelysin; K01394), matrix metalloproteinase-7 (also known as matrilysin; K01397), and matrix metalloproteinase-12 (also known as macrophage elastase; K01413). The Entrez ID-specific GO terms show that they have identical functional annotations in nearly all cases, for example the details in EcoOmicsDB for Entrez IDs 109569210 (*MMP12*, zebu cattle), 102388711 (*MMP7*, Chinese alligator), and 103089307 (*MMP3*, Yangtze river dolphin). Taken together, there is ample evidence that these differentially expressed orthologs are robust and represent real genes/proteins.

The original publication reported 405 transcripts that were significantly impacted in all three species. We quantified all RNA-seq samples with Seq2Fun, even though there was a reference transcriptome for one species. This procedure greatly simplified the downstream analysis. Since all quantified samples shared the same set of Seq2Fun IDs, the data could be integrated across species and analyzed in a single DEA, which significantly improved the statistical power because the overall sample size was 18, instead of three separate analyses with six samples each. This likely explains our 2780 DEGs versus their 405. It is possible that reads from the same gene mapped to different ortholog IDs for different salamander species, inflating the number of DEGs. To check for this, we analyzed how often similar orthologs were both differentially expressed and had distinct species-specific expression patterns. Our analysis showed that these 'duplicated' DEGs are not common, accounting for ~ 3% of the significant results. More details on this analysis are given in the Supplementary Discussion.

Finally, we note that the use of salamanders makes this a particularly strong case for our proposed RNA-seq analysis framework. Amphibians can have notoriously large genomes[27], estimated to range from 14 to 120 GB across salamander species (for reference, the human genome is 3.2 GB)[28]. Performing de novo assembly of two salamander genomes is extremely computationally intensive. The full analysis for this case study, including raw reads processing, statistical analysis, and figure preparation, took less than 24 h without using the command line or R programming, and was all done on a laptop computer. In cases where more detailed isoform analysis is desired, targeted assembly of reads mapping to individual ortholog groups of interest can be performed.

## Discussion
The main motivation of this work was to address the major computational bottlenecks facing researchers collecting RNA-seq data from non-model species[29,30]. As the costs of acquiring these data continue to drop, data analysis is increasingly becoming the bottleneck as many research groups do not have the in-house expertise or resources to process, analyze, and interpret RNA-seq data. From firsthand experience, existing point-and-click software for de novo assembly can cost upwards of $10 000 USD for the assembly and annotation alone. Hiring a contract bioinformatician is also expensive and can result in a lengthy analysis. While the initial analysis may be conducted quickly, each follow-up research query or visualization modification requires communication, typically over email. By removing barriers related to computing resources, programming skills, and knowledge of bioinformatics databases, we believe that ExpressAnalyst will make RNA-seq data processing and analysis more accessible to researchers working with non-model organisms.

Using our current framework, RNA-seq data can be easily compared across a large number of species without increasing the complexity of finding ortholog matches because reads from all species are mapped to the same ID space. This contrasts with typical BLAST-based approaches where pairwise sequence-similarity searches must be conducted between transcriptomes of different species, greatly increasing the complexity of the analysis with each additional species. One limitation of this approach is that it's possible for reads from the same gene to be mapped to similar but different orthologs for different species, and care must be taken while interpreting the results, especially when comparing more taxonomically diverse species (Supplementary Note 3). Resolving conflicting ortholog annotations is unavoidable in RNA-seq analysis of non-model organisms: even when researchers choose to analyze their data with a de novo transcriptome, they still must annotate this de novo transcriptome by drawing on functional information from other species[7,31].

EcoOmicsDB comprehensively addresses ortholog grouping and annotation across many species. When used as the database for Seq2Fun, it produces count tables that always use the same set of IDs, suitable for cross-species comparisons of transcriptomics data. Such

comparisons are of great interest as demonstrated by the recent efforts to use 'omics data for cross-species extrapolation of toxicity mechanisms and regulatory applications in the field of environmental toxicology[32]. ExpressAnalyst enables fast hypothesis-analysis-conclusion cycles through its interactive visual analytics tools such as the heatmap, ridgeline, and upset plots. By connecting dense statistical results to powerful bioinformatics databases containing functional information in a flexible and visual format, users are guided to move from overall trends and patterns to investigate prominent results in great depth.

The genomes of non-model organisms have been much less studied compared to mammalian model organisms and there are still many uncharacterized IDs in published reference genomes. With continued development and engagement with researchers working with non-model organisms, particularly on expanding Seq2Fun to quantify non-coding genes and further improving ortholog annotation, EcoOmicsDB can serve as the reference resource for transcript identification and functional annotation in non-model organisms. In future versions, we envision a system in which functional information learned from individual transcriptomics studies is added to the ortholog profiles in EcoOmicsDB. Over time, knowledge from such studies can be pooled across species to gain insights into uncharacterized proteins. This will not only improve the annotation and functional analysis of the current studies but would also benefit the whole research community that works on orthologs and non-model organism transcriptomics.

## Methods

### Overview of the Seq2Fun algorithm

Seq2Fun is designed to efficiently perform translated search of RNA-seq reads against a protein database[13]. The core algorithm is based on Burrows-Wheeler Transform (BWT) and Full-text Minute-space Index (FMI). BWT compresses millions of protein sequences to significantly reduce the size of the database while maintaining a small memory footprint. FMI creates a searchable data structure based on the compressed sequences that can quickly find matches to short amino acid sequences.

The first step in the Seq2Fun algorithm is to conduct reads quality control and join paired-end reads. Next, each cleaned read is translated into dozens of peptide sequences using all possible reading frames. In addition to finding the true peptide among these, Seq2Fun must also consider many other sequences that capture amino acid substitutions and deletions because the reference database contains many species that cover a wide taxonomic range[10,11]. Considering all possible amino acid substitutions for all possible reading frames is not computationally feasible, and so Seq2Fun employs two filters to greatly reduce the search space: peptide length and BLOSUM62 score, a widely used scoring matrix that summarizes the frequency of specific amino acids sequences in known peptides. First, translated sequences generated by different reading frames are filtered based on length based on the theory that the 'true' translation will result in a long amino acid sequence. If there are multiple amino acid sequences tied for the longest length, the most promising one is selected based on BLOSUM62 scores.

Next, the search starts with a seed of seven amino acids and considers the C terminal of the fragment to find potential start positions of matches to sequences in the reference database. A backwards searching strategy is used to extend potential matches from the C terminal to the N terminal. Allowing mismatches helps overcome evolutionary distances between the query and target sequences, however considering all amino acid substitutions would significantly slow down the search. Seq2Fun reduces the search space by again using BLOSUM62 scores to prioritize the most likely substitutions. Searches for sequences matching the query fragment continue until the N terminal is reached, or the maximum number of substitutions is exceeded. The searching regime for this read will stop if the potential match meets certain thresholds, including minimum matching scores and maximum number of substitutions. Otherwise, the search will continue based on the next most likely peptide according to the previously outlined criteria. Finally, after each read has been searched, Seq2Fun summarizes the number of best matches to each ortholog group into a count table.

We have updated Seq2Fun to version 2.0 to reduce its memory footprint (e.g., from 1.49 GB to 0.94 GB) by converting all sequences and annotations from 'string' to 'integer' data types and improving the database indexing method so that the whole process can be run in a personal computer. We notice the main bottleneck is related to input/output (I/O) speed. Adding more threads (higher than 16) may not improve performance due to this I/O constraint. Using a high-speed drive could relieve the constraint. Version 2.0 also contains a new function called SeqTract to retrieve the mapped reads for a given list of genes for transcript assembly. It takes a list of Seq2Fun IDs and the mapped fastq.gz files as an input and outputs a fastq.gz file for each ID which can be fed into popular de novo assemblers[21]. SeqTract is highly efficient supporting multi-threading with a consistent and low memory footprint.

### Creation of a high-resolution ortholog database - EcoOmicsDB

All protein-coding genes ($n = 12,828,537$) from 687 organisms covering all major phyla of eukaryotes were downloaded from KEGG using KEGGREST (version 1.34.0)[14]. Protein FASTA files for each species were submitted to OrthoFinder (version 2.5.4) for classification of genes into ortholog groups (parameters: $t = 56$, $a = 25$)[17]. OrthoFinder is a highly accurate and scalable pipeline for ortholog inference. It takes protein sequences as input and identifies all homologs by exploring both heuristic analysis of similarity scores from pairwise sequence comparison and phylogenetic trees of genes to clarify the relationship of ortholog and paralog.

The number of sequences in each ortholog group follows a power law (Fig. 2A), with the largest groups combining tens of thousands of sequences. This level of summary does not approximate gene-level count tables because tens of distinct functional groups are collapsed into one, and therefore is difficult to interpret. To solve this, we applied an additional algorithm to the top 10,000 largest ortholog groups and split them into multiple sub-groups to increase the resolution. First, MAFFT (version 7.471) was used to conduct protein alignment for each ortholog group[33]. Subsequently, the FastTree was used to generate a phylogenetic tree of all sequences in the ortholog group[34]. Next, the phylogenetic tree, or dendrogram, was converted into a distance matrix based on the pairwise cophenetic distance, which is the height of the dendrogram at the first point where two branches containing both sequences merge. Next, k-means clustering was used to split the sequences into groups based on the distance matrix. Choosing an appropriate value for $k$ for each ortholog group was challenging because we were trying to minimize the grouping of distinct functional proteins, while simultaneously maximizing the grouping of the same functional proteins across species. There is an inherent tradeoff between these two objectives because the higher the resolution (increased value of $k$), the easier to minimize grouping of functional proteins but the harder to group across species. After several iterations of clustering and evaluating the results, we defined $k$ as two times the ratio of the number of sequences to the number of species. For each ortholog group, gene-specific information was collapsed to a single gene symbol with associated text description, KEGG pathway, and GO annotation by tabulating the frequency of symbols and descriptions and choosing the most frequent ones after removing generic terms such as "uncharacterized" based on manual inspection.

EcoOmicsDB currently consists of 29 taxonomic sub-group databases based on the NCBI taxonomy system (Table 1). Selecting an appropriate sub-group database can improve performance in terms of

speed and specificity for reads annotation and quantification. However, if the protein database is not selected properly (e.g., reads from a protist sample are mapped to the "fishes" database), key functional groups may be missed. As an empirical guideline for quality control, we provide summary statistics on the set of "core" orthologs for each sub-group (defined as orthologs that are present in the genomes of > 90% of sub-group species) and advise that > 80% of core genes should be typically quantified when a taxonomically appropriate database is selected.

### Implementation of the web-based platforms

ExpressAnalyst is implemented based on PrimeFaces and PrimeNG (www.primefaces.org) libraries (version 12 and 13, respectively) and R (version 4.1.3). The raw data processing module performs reads quality check based on fastp (version 0.21.1), and quantification based on Kallisto (version 0.46.1) or Seq2Fun (version 2.0.2). For the public server, FASTQ file upload is handled with FileBrowser (version 1.3.6), and jobs are managed by Slurm (version 20.11.2). Users frequently reported that data uploading is the most time-consuming and challenging part, due to limited bandwidth or data security concerns. To address this, we have created a Docker image of the raw data processing module to enable quantification on a local computer through our user-friendly web interface. Many features of ExpressAnalyst were previously published as part of the NetworkAnalyst platform[35]. Here, we split the general gene expression profiling features from the network building and visualization features. Significant efforts were made to develop high-performance interactive volcano plot, heatmaps and ridgeline plot to facilitate exploratory data analysis, with built-in annotation and functional analysis options to support count tables produced by Kallisto or Seq2Fun. ExpressAnalyst is available at www.expressanalyst.ca.

### Case study methods

Zebrafish RNA-seq data were obtained from NCBI's Sequence Read Archive (SRA) with sample accessions SRR13332314 – SRR13332325 (https://trace.ncbi.nlm.nih.gov/Traces/?view=study&acc=SRP299836). Files were downloaded and converted to FASTQ format using the NCBI SRA ToolKit (version 2.11.3) before being uploaded to ExpressAnalyst. Details on sex were not included in the original manuscript. Briefly, stage I lobster larvae were exposed for 24 h to four doses of WAF (10%, 19%, 37%, and 72%) a positive control (1-methylnaphthalene, at 0.3 mg/L corresponding to the estimated concentration of EC20) or a negative control (0.22 μm filtered seawater). Due to the lack of effects reported on survival, molting and respiration after WAF exposures, only the highest dose of WAF (72% WAF) was considered for transcriptomics analysis. RNA extraction of whole larvae exposed to 72% WAF ($n = 7$), methylnaphthalene ($n = 6$) and filtered seawater ($n = 6$) was performed using Trizol. The transcriptomes were sequenced using Novaseq Illumina at 28 M reads per library. FASTQ files were downloaded from the Genome Quebec portal and uploaded to ExpressAnalyst. Sex was not considered as part of the experimental design due to the logistical difficulty of sexing invertebrate larvae. Ethical approval was not required since invertebrates other than cephalopods are not regulated under the Canadian Council on Animal Care guidelines.

For the Kallisto workflow, zebrafish samples were aligned to the GRCz11 *Danio rerio* genome assembly (accession: GCA_000002035.4, https://www.ncbi.nlm.nih.gov/data-hub/genome/GCF_000002035.6/) and lobster samples were uploaded to the GMGI_Hamer_2.0 *Homarus americanus* genome assembly (accession: GCA_018991925.1, https://www.ncbi.nlm.nih.gov/data-hub/genome/GCF_018991925.1/). The minimum quality score parameter was set to 25. For the Seq2Fun workflow, zebrafish samples were aligned to a customized version of the "Fishes" database that had zebrafish sequences removed, and lobster samples were aligned to the 'Crustaceans' database. The Seq2Fun parameters were set as "maximum number of

mismatches" = 2, "minimum matching length" = 19, and "minimum matching BLOSUM62 score" = 80. The zebrafish sequences were removed to better demonstrate how mapping to pooled sequences from many other species compares to mapping to a reference transcriptome. This customized database is available for download in the Seq2Fun website (www.seq2fun.ca) under the "Database" tab.

Both count tables (Kallisto and Seq2Fun) were analyzed with ExpressAnalyst. The appropriate organisms and IDs were selected, and data type was set to RNA-seq. Data were filtered to remove those with low abundance and low variation by setting the variance filter to 15 and the abundance filter to 4. Data were normalized using the "Log2 counts per million" option followed by differential analysis with the limma R package (version 3.52.4). Each treatment group was compared to the control group using the "specific comparison" option. Genes were defined as differentially expressed if their false discovery rate (FDR) adjusted p-values were less than 0.05 and the absolute log fold changes greater than 1.5. For each contrast, the list of DEGs was analyzed for enriched KEGG pathways using the "ORA Networks" tool. A pathway was defined as significantly enriched if the FDR-adjusted p-value was less than 0.05 and there were at least five DEGs in the gene set.

Salamander RNA-seq data were obtained from NCBI's SRA at sample accessions SRR7499348 - SRR7499365, excluding sample SRR7499350 which was identified as an outlier by QA/QC performed by the original publication[25]. All RNA-seq profiles were measured from distinct tissue samples. Files were downloaded and converted from SRA to FASTQ format using the NCBI SRA ToolKit (version 2.11.3) before being uploaded to ExpressAnalyst. Samples were aligned to the "Vertebrates" database using Seq2Fun 2.0. In the 'Expression Profiling' module, the count table was normalized with the "Log2 counts per million" option using the limma voom R package followed by differential analysis with limma[23]. For differential analysis, time was set as the primary factor, and species as the secondary factor with the secondary factor defined as a "blocking factor". Then, a "specific comparison" was performed between 'time0' and 'time24'. Following the original publication[25], genes were considered differentially expressed if their FDR-adjusted p-values were less than 0.1. The DEGs were split into lists of up and down-regulated genes, and each list was analyzed for enriched KEGG pathways using the "Interactive Volcano Plot" tool, which uses hypergeometric tests to perform over-representation analysis. A pathway was defined as significantly enriched if the FDR-adjusted p-value was less than 0.05 and there were at least five DEGs in the gene set.

### Reporting summary

Further information on research design is available in the Nature Portfolio Reporting Summary linked to this article.

## Data availability

The salamander RNA-sequencing data re-analyzed in this study were deposited by the original authors in the NCBI Sequence Read Archive under accession code SRP152819. The lobster RNA-sequencing data generated in this study have been deposited in the NCBI Gene Expression Omnibus database under accession code GSE225876. The zebrafish RNA-sequencing dataset re-analyzed in this study were deposited by the original authors in the NCBI Sequence Read Archive under accession code SRP299836. EcoOmicsDB is freely available for bulk download at www.ecoomicsdb.ca/#/download. All findings in the manuscript can be reproduced using these data.

## Code availability

Source code for the Seq2Fun algorithm are available at https://github.com/xia-lab/Seq2Fun[36]. R code for the ExpressAnalyst statistical modules are available at https://github.com/xia-lab/ExpressAnalystR[37].

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

## Acknowledgements

This project was funded by Genome Canada and Génome Québec's Bioinformatics and Computational Biology (B/CB) grant to NB and JX. We thank the team at the Huntsman Marine Science Centre for providing lobster data through funding from the Government of Canada's Department of Fisheries and Oceans Multi-Partner Research Initiative. We thank early users who provided valuable feedback. We also thank Joseph O'Brien for technical assistance.

## Author contributions

The overarching goals and ideas of this project were conceptualized by P.L., J.E., E.L., Y.S.J., J.S., J.H., N.B., and J.X.; The novel statistical methods were developed by P.L., J.E., and J.X.; Formal analysis, data curation, and visualization were done by J.E. and P.L.; P.L., J.E., O.H., Z.P., G.Z., and J.X. contributed to the development of software. The software was validated and improved by general user feedback from J.E., O.H., E.L., Y.S.J., J.S., J.H., and N.B.; J.H. and J.X. contributed resources, including samples and computing resources. The original draft was written by P.L., J.E., and J.X., and all authors contributed to reviewing and editing of the final manuscript. N.B. and J.X. acquired funding and supervised the project.

## Competing interests

The authors declare the following competing interests: J.E., Z.P., G.Z., and J.X. own shares of OmicSquare Analytics Inc. The remaining authors declare no competing interests.
