## [Peer Review File · Nature Communications]

ExpressAnalyst: a unified platform for RNA-sequencing analysis in non-model speciesREVIEWER COMMENTS

Reviewer #1 (Remarks to the Author):

Transcriptomic studies of non-model organisms are accumulating in recent years; however, due to lack of high-quality reference genomes and transcriptomes, the computational analysis of such data remains challenging, especially for those without advanced bioinformatics skills. To solve this problem, Liu et al. presented a unified web-based suite composed of three modules, including EcoOmicsDB for ortholog mapping, EcoOmicsAnalyst for data processing and annotation, and ExpressAnalyst for statistical and functional analysis.

Given that several involved core tools (eg. Seq2Fun INMEX and INVEX) have already been reported by the same group previously (Liu et al., Genome Res, 2021; Xia et al., Nucleic Acids Res, 2013; Xia et al., Bioinformatics, 2014), the presented algorithms and tools are not completely novel. The merit of this study is that the authors did make great efforts to optimize the algorithm and extend the annotation database, and they finally integrated multiple modules as an easy-to-use unified platform which can be easily adopted by users, even for those lack advanced bioinformatics skills. In addition, they also developed several new modules (such as the tools for expression profiling in the ExpressAnalyst module) to facilitate data analysis and visualization.

Overall, the paper is well written, and the software suite is well implemented. By offering an easy-to-use and all-purposed solution, I expect this study could significantly facilitate the study of the RNA-Seq data of non-model organisms.

I have several comments and suggestions for further improvement.

1. The three modules are developed to perform several related tasks, yet currently each module is represented in separated webpage, which makes the switching between them inconvenient. The authors are highly encouraged to create an integrated webpage to include all modules. Ideally, the authors can enable the data transfer from one module to another, or at least they could create a single webpage to include the URLs and descriptions for all modules.

2. I failed to access EcoOmicsDB using FileZilla following the instruction from: <https://omicsforum.ca/t/how-can-i-upload-my-raw-rna-seq-data-files/637>. The login was succeeded, but then we got the error "Failed to retrieve directory listing". This precludes further testing of this module. The remaining two modules work properly.

3. Some technical details are missed. For example, how to achieve efficient ortholog matching (line 390)? How to reduce memory use (line 459)?

4. For all the figures and tables, it is suggested to include the titles apart from the legends. Also, the brackets for A), B), C) etc in the figures can be removed.
5. The “Results” and “Methods” sections are a little lengthy and sometimes redundant. For example, in page 11, the detailed instruction about EcoOmicsAnalyst can be provided as supplemental documents or shown in the website. Page 23 and 24, the authors can just highlight the changes and improvements in version 2.0, since Seq2Fun has already been published before. Page 25 line 480, the sentence related to OrthoFinder can be removed.
6. In Table 1, what does the last column named Seq2Fun(v1) stands for? Also, the sum of species is 687 but not 596.
7. Line 291, regarding the analysis of data from GEO database, it would be helpful to enable SRA to FASTQ conversion in the webserver.
8. Line 295, regarding the comparison across species, how the count matrix is normalized? Did the authors consider gene length, which may differ across species and influence expression quantification?
9. Line 388, I assume the developed tools would have broader application than those related to environmental life sciences.
10. A few typos need to be corrected, such as: Line 110 “with” should be “without”; Line 167, “toxicology-focused authors” is confusing; Line 335, “provides” should be “provide”; Line 546, “was” should be “were”
11. Line 593-596, the two citations for Holzer et al. are duplicated.

Reviewer #2 (Remarks to the Author):

This paper introduces a web-based platform for performing RNA-seq analysis. Whereas there are many tools for transcriptome analysis for human and other organisms with a well-annotated reference genome, this platform aims at non-model organisms without a reference genome and probably fills a gap. The main scientific contribution of the group, and a core of this platform, is the previously published Seq2fun program that aligns RNA-seq reads against a protein database. Here an updated version of this tool is presented, which faster and use less memory. The main contribution of this work is the creation of a data base of orthologous proteins and integration with Seq2fun and other tools into an end-to-end analysis platform. They have put a great deal of effort into creating a user-friendly platform that can be easily used by non-bioinformaticians. The paper is very well written.

My only concern is the case study that I find unconvincing. They find almost 3000 DEGs, which is probably half the expected number of expressed genes, which makes an analysis almost useless for further biological understanding. Although it may be a correct result, it does raise some suspicion of false positives resulting from e.g. the same gene in different species mapping to different orthologs or

phenomena similar to that. The authors need to discuss sources of errors and extend the analysis to address this. Note that it is not enough to test with data from a species with a reference genome, unless that and very closely related genomes are removed from the database of orthologs.

Please see below our response to the comments from Reviewers 1 and 2:

Comments from Reviewer 1:

1. The three modules are developed to perform several related tasks, yet currently each module is represented in separated webpage, which makes the switching between them inconvenient. The authors are highly encouraged to create an integrated webpage to include all modules. Ideally, the authors can enable the data transfer from one module to another, or at least they could create a single webpage to include the URLs and descriptions for all modules.

Our response:

Thank you for the advice. We agree that the three websites are confusing for users. Based on the suggestion, we have merged all modules under the ExpressAnalyst platform (www.expressanalyst.ca).

In addition, the output of the raw data processing module is pre-formatted for statistical analysis with the 'table' module in ExpressAnalyst, and links are enabled between the two tools. The main differential expression results table is linked to EcoOmicsDB (<https://expressanalyst.ca/EcoOmicsDB>) when analyzing data quantified with Seq2Fun.

2. I failed to access EcoOmicsDB using FileZilla following the instruction from: <https://omicsforum.ca/t/how-can-i-upload-my-raw-rna-seq-data-files/637>. The login was succeeded, but then we got the error "Failed to retrieve directory listing". This precludes further testing of this module. The remaining two modules work properly.

Our response:

McGill University recently updated their firewall settings, making it impossible for most users outside the McGill network to access our server via FTP. To solve this issue, we have switched from using FileZilla to a much more modern technology based on FileBrowser (<https://filebrowser.org>). Now, a user does not have to install any additional software - file upload is a simple 'drag and drop' operation.

We have also worked to make our Docker implementation more user friendly to encourage local processing for larger datasets. We have worked closely with several bench scientists with limited programming experience try to run our Docker, updating our documentation and interface to clarify pain points, and then repeating until the process was consistently smooth. In addition, we have significantly improved our tutorials on step-by-step instructions and troubleshooting tips.

3. Some technical details are missed. For example, how to achieve efficient ortholog matching (line 390)? How to reduce memory use (line 459)?

Our response:

Thank you for the comment. We have added the following details:

L390: “Using our current framework, RNA-seq data can be easily compared across a large number of species without increasing the complexity of finding ortholog matches because reads from all species are mapped to the same ID space.”

L459: “We have updated Seq2Fun to version 2.0 to reduce its memory footprint (e.g., from 1.49 GB to 0.94 GB) by converting all sequences and annotations from ‘string’ to ‘integer’ data types and improving the database indexing method so that the whole process can be run in a personal computer.”

4. For all the figures and tables, it is suggested to include the titles apart from the legends. Also, the brackets for A), B), C) etc in the figures can be removed.

Our response:

We have updated the figures and tables. The brackets have been removed from all figures.

5. The “Results” and “Methods” sections are a little lengthy and sometimes redundant. For example, in page 11, the detailed instruction about EcoOmicsAnalyst can be provided as supplemental documents or shown in the website. Page 23 and 24, the authors can just highlight the changes and improvements in version 2.0, since Seq2Fun has already been published before. Page 25 line 480, the sentence related to OrthoFinder can be removed.

Our response:

Thank you for the suggestion. We have moved the description of the raw data processing module and the Seq2Fun algorithm to the supplemental materials and the sentences starting on L480 were removed, reducing the length of the main manuscript file by ~2 pages.

6. In Table 1, what does the last column named Seq2Fun(v1) stands for? Also, the sum of species is 687 but not 596.

Our response:

We apologize, the 596 species were left over from a different version of EcoOmicsDB. We have verified all species numbers and changed 596 to 687 throughout the manuscript. The last column of EcoOmicsDB shows the number of orthologs in the original KEGG Ortholog database used for the original publication and in Seq2Fun version 1. It shows the vast improvement in transcriptome coverage using our custom ortholog definition method. We updated the Table 1 legend to include a brief explanation of this column.

7. Line 291, regarding the analysis of data from GEO database, it would be helpful to enable SRA to FASTQ conversion in the webserver.

Our response:

Thank you for the suggestion. In our experience, this process is quite memory intensive, which is hard to maintain and sustain for concurrent users in a public web server. We are going to add this support for our Docker version in the future.

8. Line 295, regarding the comparison across species, how the count matrix is normalized? Did the authors consider gene length, which may differ across species and influence expression quantification?

Our response:

Thank you for the comment. We point the reviewer to the case study methods section, where we specify that the data were “normalized with the “Log2 counts per million” option using the limma voom R package”. This method does not account for gene length, however the primary goal of the analysis is to identify differentially expressed genes where the treatment groups are always compared to a within-species control, and not to directly compare expression levels across different genes (where gene length must be accounted for). Additionally, species is included as a random effect (blocking factor in limma terminology) in the linear models used for differential analysis, allowing us to account for systematic differences related to species when identifying genes that are regulated in the same direction across species. Accounting for gene length when the reference database is a compilation of sequences from hundreds of species is an interesting problem. We agree that this could be useful for analyses that compare expression levels between genes, and will add it to the list of features to be implemented in version 2.0 of ExpressAnalyst.

9. Line 388, I assume the developed tools would have broader application than those related to environmental life sciences.

Our response:

We adjusted the phrasing to refer to non-model organisms instead of to the environmental life sciences: “will make RNA-seq data processing and analysis more accessible to researchers working with non-model organisms”.

10. A few typos need to be corrected, such as: Line 110 “with” should be “without”; Line 167, “toxicology-focused authors” is confusing; Line 335, “provides” should be “provide”; Line 546, “was” should be “were”

Our response:

Thanks for pointing out these typos, we have corrected them.

11. Line 593-596, the two citations for Holzer et al. are duplicated.

Our response:

Thanks for pointing this out, we have corrected this.

Comments from Reviewer 2:

My only concern is the case study that I find unconvincing. They find almost 3000 DEGs, which is probably half the expected number of expressed genes, which makes an analysis almost useless for

further biological understanding. Although it may be a correct result, it does raise some suspicion of false positives resulting from e.g. the same gene in different species mapping to different orthologs or phenomena similar to that. The authors need to discuss sources of errors and extend the analysis to address this. Note that it is not enough to test with data from a species with a reference genome, unless that and very closely related genomes are removed from the database of orthologs.

Our response:

We appreciate this question - we have debated a lot regarding this finding within our bioinformatics team as well as with our collaborators who are bench scientists with experience on this area. We provide our answers from several perspectives.

1) Analysis context

We agree that 3000 DEGs is a lot, however there were 20,687 orthologs quantified so it is ~14% not 50%, a more reasonable percentage. Additionally, we kept the same thresholds as the original publication to keep overall comparisons of the functional results as consistent as possible. Please note the original publication used an FDR threshold of 0.1 instead of the conventional 0.05. When we use an FDR threshold of 0.05, the number of DEGs is decreased to 1926 (~9% of all quantified orthologs). We also note that limb amputation is an extreme experimental condition where we expect a strong transcriptomic effect compared to more subtle experimental conditions such as variable exercise, diet, temperature, or chemical exposure.

2) Database composition

Inflated DEG lists caused by the same gene being mapped to different orthologs for different species is a worthwhile problem to investigate. We do not expect it to be a substantial factor in our case study because there are only three amphibian species in the Seq2Fun database, none of which are salamanders, and so the chances of the same functional gene being mapped to different ortholog groups is low given the taxonomic resolution of our species versus the species in the ortholog database.

3) Database evaluation

We did further analysis to investigate how many of our DEGs could be 'duplicated', or in other words, the result of the same gene being mapped to different ortholog groups for different species and being detected as differentially expressed with respect to time in both cases. Our ortholog ID system is partly hierarchical – during definition, we first identified large and general orthologs, and then split these into more refined groups using an adaptive k-means clustering approach (see methods section for more details). We leveraged this hierarchical organization to assess how often the same general ortholog group both A) contained specific ortholog IDs that were lowly expressed in some species and robustly expressed in others, and B) contributed multiple DEGs to the case study results.

To do this, we first computed the average counts for each ortholog group for each species. Next, we identified cases where the average counts were very low for one or two species (mean counts < 2) AND robustly expressed for one or two species (mean counts > 10). Next, we annotated each of these specific IDs with its more general ortholog group ID, and noted whether it was identified as differentially expressed in our case study. Finally, we counted the number of case study DEGs that were part of a general ortholog group that had multiple DEGs with distinct species expression patterns. We found that

83 DEGs (~3%) fit both these criteria. While this is an approximation of the type of false positives the reviewer is referring to, we believe that it shows that this problem is not a great concern for this particular case study.

We added a section in the Supplementary Information detailing this analysis, and briefly reference it in the main text:

“It is possible that reads from the same gene mapped to different ortholog IDs for different salamander species, inflating the number of DEGs. To check for this, we analyzed how often similar orthologs were both differentially expressed and had distinct species-specific expression patterns. Our analysis showed that these ‘duplicated’ DEGs are not common, accounting for ~ 3% of the significant results. More details on this analysis are given in the Supplementary Discussion.”

This could be a problem with cross-species analyses where the species are taxonomically very different from each other, for example birds vs. fish vs. mammals, and we acknowledged this in the section of the discussion on using Seq2Fun for cross-species comparisons:

“One limitation of this approach is that it’s possible for reads from the same gene to be mapped to similar but different orthologs for different species, and care must be taken while interpreting the results, especially when comparing more taxonomically diverse species.”

Finally, we note that for the other two case studies, American lobster transcripts were not included in the Seq2Fun database and a custom fish database without *Danio rerio* sequences was created specifically for this analysis. We have added this detail to the description in the supplementary information:

“For the Seq2Fun workflow, zebrafish samples were aligned to a customized version of the “Fishes” database that had zebrafish sequences removed”

These reference transcriptome:Seq2Fun database comparisons mirror the in-depth validation exercises that we performed in our original publication, where we showed that mapping short reads directly to a collection of orthologous sequences from other species is more accurate, stable, and computationally efficient approach than *de novo* transcriptome assembly. In this manuscript, the objective was to provide a quick ‘sanity check’ that swapping out one ortholog database for another does not fundamentally change how the algorithm works or the variability structure of the quantified counts, it simply presents the results in more refined detail compared to version 1.0.

REVIEWERS' COMMENTS

Reviewer #1 (Remarks to the Author):

I am glad that the authors have made great efforts to improve their work. During the revision, they have merged all modules to a single platform, corrected the bugs for data upload, and improved the Docker for more flexible analysis. They have also provided more explanations for the technical details, and further polished the writing throughout the manuscript. Overall, the improvement after the revision is very impressive.

I just have a few minor comments for further improvement:

1. The titles are still missed for all main text figures and tables. Apart from providing the explanation for each panel, it is highly encouraged to provide a title to summarize the entire figure.
2. A few typos need to be corrected, for example: Line 33, "raw data processing and annotation of FASTQ files" is confusing since FASTQ files and raw data usually mean the same thing; Line 34, "counts tables" should be "count tables".

Reviewer #2 (Remarks to the Author):

I thank the authors for the detailed response to the points I raised and in particular the additional analysis of possible sources of false positives. I acknowledge that I underestimated the "expected number of expressed genes" in my review not realizing that the samples probably contain a mixture of many different cell types. I am satisfied with the revised manuscript.

Please see below our response to the comments from Reviewer 1:

Comments from Reviewer 1:

I am glad that the authors have made great efforts to improve their work. During the revision, they have merged all modules to a single platform, corrected the bugs for data upload, and improved the Docker for more flexible analysis. They have also provided more explanations for the technical details, and further polished the writing throughout the manuscript. Overall, the improvement after the revision is very impressive.

I just have a few minor comments for further improvement:

1. The titles are still missed for all main text figures and tables. Apart from providing the explanation for each panel, it is highly encouraged to provide a title to summarize the entire figure.

Our response:

Thank you for the clarification, we did not understand your comment during the previous review. We have added titles to all figures and tables.

2. A few typos need to be corrected, for example: Line 33, “raw data processing and annotation of FASTQ files” is confusing since FASTQ files and raw data usually mean the same thing; Line 34, “counts tables” should be “count tables”.

Our response:

Thanks, we have corrected these typos. We additionally corrected several other instances where we had said “counts tables”.